# Social Support—A Protective Factor for Depressed Perinatal Women?

**DOI:** 10.3390/ijerph16081426

**Published:** 2019-04-21

**Authors:** Jeannette Milgrom, Yafit Hirshler, John Reece, Charlene Holt, Alan W. Gemmill

**Affiliations:** 1Parent-Infant Research Institute, Heidelberg West, Victoria 3081, Australia; Jeannette.MILGROM@austin.org.au (J.M.); Yafit.HIRSHLER@austin.org.au (Y.H.); Charlene.HOLT@austin.org.au (C.H.); 2Melbourne School of Psychological Sciences, University of Melbourne, Melbourne, Victoria 3010, Australia; 3School of Psychological Sciences, Australian College of Applied Psychology, Melbourne, Victoria 3000, Australia; John.Reece@acap.edu.au

**Keywords:** antenatal depression, antenatal anxiety, postnatal depression, postnatal anxiety, parenting stress, social support, child development

## Abstract

Social support before and after childbirth is a possible protective factor for perinatal depression. Currently, there is a lack of longitudinal studies beyond the first year postpartum exploring the relationship of social support with depression and anxiety. Social support is also a possible protective factor for adverse child development, which is a known consequence of perinatal depression. The present study followed up a cohort of depressed women (*n* = 54) from a randomised controlled trial of psychological treatment for antenatal depression. We examined the trajectory of the relationships between perceived social support (Social Provisions Scale), depression (Beck Depression Inventory), and anxiety (Beck Anxiety Inventory) twice in pregnancy and twice postpartum up to two years. The influence of social support on child development and parenting-related stress was also explored. Two aspects of social support, Reassurance of Worth and Reliable Alliance, were strongly related to perinatal depression and anxiety, particularly when predicting symptoms in late pregnancy. However, the effect of postnatal depression on child development at 9 and 24 months post-birth was not mediated by social support. These results suggest the importance of adjusting current interventions for depressed perinatal women to focus on social support in late pregnancy and the first six months postpartum.

## 1. Introduction

Social support has been shown to play a protective role against depression in the general population [1,2,3]. The role of social support in the perinatal period has been of particular interest as a possible protective factor for coping difficulties arising from the many challenges that motherhood brings [4,5,6,7]. 

Previous studies have pointed to a link between social support and perinatal depression. In pregnant women, depression has been found to be inversely associated with social support; namely women with lower social support report more depression symptoms than women with higher social support [8,9,10]. Postnatal depression is also inversely associated with social support [11,12,13,14]. The majority of studies have been cross-sectional, with only a few longitudinal studies demonstrating a relationship between social support and depressive symptoms. For example, Leahy-Warren and colleagues [15] followed postnatal women to three months postpartum, and Brown and colleagues [16] followed postnatal women to one-year postpartum, although the latter study found that social support played a role only for women with higher baseline depressive symptoms. To date, it is still unclear whether this association persists beyond the first year postpartum, and which different aspects of social support may be important during pregnancy, post-birth, and early infancy. 

As social support is a multidimensional construct, various scales have been developed to reflect functional and structural dimensions. Seguin and colleagues [17] described five aspects of functional support: Instrumental, emotional, informational, positive feedback, and companionship. Cutrona and Russell [18] developed the social provision scale (SPS) with six dimensions, which greatly overlap with Seguin’s functional dimensions, namely attachment (parallels with emotional), social integration (parallels with companionship), reassurance of worth (parallels with positive feedback), reliable alliance (parellels with instrumental support), guidance (parellels with informational support), and opportunity for nurturance. The additional item in Cutrona’s scale (i.e., opportunity for nurturance) seems very pertinent to the mother–infant relationship [19], as this dimension refers to responsibilty for the wellbeing of another individual. 

As different aspects of support may be needed in different contexts [17,20,21], it is of interest to evaluate each of them separately from pregnancy over the first two years postpartum. A number of interesting findings have emerged to date. For example, at three weeks postpartum, unavailability of informational support [17] and lack of instrumental support [20] increased the risk for depressive symptoms. At eight weeks postpartum, inadequate social integration during pregnancy increased the risk of depressive symptoms [21]. 

The role of social support as a possible protective factor not only for depression but also on child developmental outcomes is of interest. One of the major consequences of perinatal depression is adverse child outcomes [22,23,24,25,26,27,28]. Whilst social support may be a possible moderating factor, few studies have explored the involvement of social support in the relationship between depression and anxiety and child development. Recently, McDonald and colleagues [29] found that social support during pregnancy was a protective factor against delayed child development but only in women with socio-demographic risk and not in women with mental health risk. 

Of the few studies conducted on this topic, the focus has been on the association between social support and birth outcomes, which, in turn, may be related to child development [30,31,32]. Women experiencing lower social support reported lower birth weight infants [33], worse labour progress, and babies with lower Apgar scores compared to women experiencing higher social support [20]. Moreover, depressed women experienceing lower social support had babies who were born earlier and provided lower Apgar scores than depressed women experiencing higher social support [34,35]. A similar buffering effect of social support on birth weight has been found in women experiencing stress [36]. More research is needed in order to determine the role of social support in the relationship between perinatal depression and child development.

The present study is a longitudinal follow-up of a previous randomised treatment trial for antenatal depression, which reported on maternal and child outcomes at two years [37]. The current study examines the relationships between perceived social support and depression twice during pregnancy and at six months and twenty four months postnatally. 

Much of the social support research in the perinatal period has focussed on links with depression. However, studies have also begun to examine the relationship between social support and perinatal anxiety [8,38] alongside depression as a measure of psychological wellbeing in the transition to parenthood [39]. In addition, we measured parenting stress, which is commonly elevated in women experiencing postnatal depression and anxiety [40,41].

The primary aim was therefore to explore the pattern of longitudinal relationships between social support and psychological wellbeing (i.e., depression, anxiety, and parental stress) in this cohort. Two main methodological approaches were employed: (1) Multivariate analyses were used to evaluate the canonical relationships between multiple measures of social support (i.e., the sub-scales of our social support measure) and our measures of psychological wellbeing (i.e., anxiety and depression). Mulitivariate multiple regression conducted at four time points was the analysis of choice here. (2) Time-lagged correlations were used to explore the possible long-term impact of social support on psychological wellbeing. For these analyses, the focus was on correlations between social support at one time point and psychological wellbeing at a subsequent point.

A secondary aim concerned the possible role of social support as a mediator of the relationship between parental psychological wellbeing and child behaviour outcomes. Partial correlation was used to provide a preliminary investigation of this relationship.

In summary, we hypothesised that social support plays a protective role against perinatal depression and anxiety, and influences the effect of antenatal and postnatal depression on child development.

## 2. Methods

### 2.1. Study Design

This study was a secondary analysis of data from a previously reported [37] parallel two-group randomised controlled trial (RCT) comparing cognitive behavioural therapy (CBT) treatment for antenatal depression and anxiety to treatment as usual (TAU). Details of recruitment, randomisation, treatment conditions, and intervention efficacy are given more fully elsewhere [37,41]. Briefly, the original RCT was configured as follows: At baseline, 54 pregnant women enrolled in the RCT, all of whom gave informed consent, were aged ≥18 years, less than thirty weeks pregnant, and met diagnostic criteria for major or minor depression or adjustment disorder with mixed depression and anxiety using the Structured Clinical Interview for the DSM-IV (SCID) [42]. Women were randomised to either the Beating the Blues before Birth treatment program, an 8-week CBT program, or to TAU, which meant that women were case-managed by their midwife or general practitioner and referred to other services/agencies as necessary, as would normally have happened in routine practice. The CBT treatment proved effective at ameliorating symptoms of depression and anxiety during pregnancy [37,41]. Strong reductions in anxiety were observed during pregnancy, and improvements in depression were maintained at nine months, representing a moderately large effect size [37]. 

### 2.2. Recruitment to 2-Year Follow-up 

The original RCT and the follow-up study have been approved by the Human Research Ethics Committees of Northern Health, Austin Health, and Mercy Health, Melbourne, Australia (Trial Registration ACTRN12607000397415).

### 2.3. Measures

Maternal measures (see below) were collected at baseline (T1), nine weeks post-randomisation (T2), and six months (T3), nine months (T4), and twenty four months (T5) post-birth, with the exception of PSI, which was only collected at twenty four months post-birth. Infant measures (see below) were only collected at nine months (T4) and twenty four months (T5) of age. 

#### 2.3.1. Participant Characteristics at Baseline 

Baseline information on depression diagnosis, age, and gestational stage were collected. In addition, relationship status, education, number of previous children, place of birth, and annual family income were also collected. 

#### 2.3.2. Maternal Measures at Baseline, 9 Weeks Post-Randomisation, 6-Month, and 2-Year Follow-Ups

##### The Beck Depression Inventory II (BDI-II)

The BDI-II [43] is a widely used, self-reported, well-validated, 21-item clinical instrument that measures cognitive, affective, and physiological factors to assess severity of depression. It includes groups of statements related to symptoms such as loss of pleasure, loss of interest, worthlessness, etc. Respondents select the one statement in each group that best describes the way they have been feeling during the past two weeks with rating between 0 and 3 reflecting increased severity. The BDI-II has been validated against gold-standard diagnostic criteria in perinatal populations [44].

##### The Beck Anxiety Inventory (BAI)

The BAI is a self-reported 21-item instrument for measuring anxiety, designed for use in conjunction with the BDI-II, and also has well-established psychometric properties [45]. It includes assessment of symptoms such as nervousness, dizziness, inability to relax, etc. Respondents indicate how much they have been bothered by each symptom over the past two weeks. Responses are rated on a 4-point Likert scale and range from 0 (not at all) to 3 (severely). The BAI has been used in previous research in perinatal populations [44]. 

##### The Social Provisions Scale (SPS)

The SPS [21] is a 24-item scale that covers the six components or "provisions" of social support identified by Weiss [19]. These six components include: (a) Attachment, intimate relationships in which the person receives a sense of security and safety; (b) social integration, a network of relationships in which individuals share interests and concerns; (c) opportunity for nurturance, relationships in which the person is responsible for the wellbeing of another; (d) reassurance of worth, relationships in which the persons’ skills and abilities are acknowledged; (e) reliable alliance, relationships in which the person can count on others for assistance under any circumstances; and (f) guidance, relationships with trustworthy and authoritative individuals who can provide advice. Respondents were asked to rate the degree to which their social relationships are currently supplying each of the provisions. 

##### The Parenting Stress Index (PSI)

The PSI [46] measures parent–child relationship functioning and attachment by parent report. Responses produce six ‘Child’: (distractibility/hyperactivity, adaptability, reinforces parent, demandingness, mood, and acceptability) and seven ‘Parent’ sub-scales (competence, isolation, attachment, health, role restriction, depression, and spouse/parenting partner relationship), that reflect child and parent characteristics that may contribute to overall stress in parents. A score for a child domain, parent domain, and a total score is then calculated. The PSI was administered at the twenty four-month follow-up only.

#### 2.3.3. Child Measures at 9-Month and 2-Year Follow-Up

##### The Revised Infant Behaviour Questionnaire Short Form (IBQ-R) 

The IBQ-R [47,48] is a parent-report measure of infant temperament assessing the following dimensions: Activity level, distress to limitations, approach, fear, duration of orienting, smiling and laughter, vocal reactivity, sadness, perceptual sensitivity, high-intensity pleasure, low-intensity pleasure, cuddliness, soothability, and falling reactivity/rate of recovery from distress. The IBQ-R was administered at the nine-month follow-up evaluation. 

##### The Ages and Stages Questionnaire (ASQ-3)

The ASQ-3 [49,50] assesses developmental progress of children under 5 years of age. Three subscales of the ASQ-3 (18 items) were used to evaluate developmental milestones in three areas: Communication, problem solving, and personal-social. The ASQ-3 was administered at the nine-month follow-up evaluation. 

##### The Ages and Stages Questionnaire: Social Emotional (ASQ-SE)

The ASQ-SE [51] assesses infants’ social and emotional behaviour in seven domains: Self-regulation, compliance, communication, adaptive functioning, autonomy, affect, and interaction with people. The ASQ-SE was administered at the nine-month follow-up evaluation. 

##### The Bayley Scales of Infant Development (BSID-III)

The BSID-III [52] was the primary measure of cognitive and motor development at 2 years. The BSID-III is the most widely-used measure of child development, and yields information in cognitive and motor domains. The test was administered and rated by a clinician. 

##### The Child Behaviour Checklist (CBCL)

The CBCL [53] Total Problems Score was the primary measure of behavioural development. The CBCL is a widely used diagnostic screening assessment, completed by caregivers, which, in this study, were the children’s mothers. The preschool version for children aged 1.5–5 years was used in this study. Seven subscale scores of the CBCL were calculated (emotional/reactivity, anxiety/depression, somatic problems, withdrawn behaviour, sleeping problems, attention problems, and aggressive behaviour), as well as internalizing and externalizing behavioural problems, total problems score, and the composite difficulties in emotional regulation Scale. 

### 2.4. Statistical Analysis

The data for this study were taken from a two-group RCT. Because the analyses in this study were designed to incorporate all participants, it was necessary to statistically control for group membership and the subsequent treatment effect on depression and anxiety. To achieve this outcome, group membership was regressed on both depression and anxiety scores at each time point, with the resultant residual values used as dependent outcomes in analysis.

The relationship between social support and depression and anxiety was analysed using multivariate multiple regression. Depression and anxiety were the observed outcomes of the underlying construct of psychological wellbeing and so were combined into a single multivariate outcome, with the sub-scales of social support as predictors. These analyses were conducted separately at four time points. If a significant multivariate effect was identified, separate regression analyses were examined for each of depression and anxiety, with the role of individual social sub-scales considered at the final stage. The analysis of the role that social support might play in the relationship between anxiety and depression and child behaviour was conducted using a combination of Pearson correlation and partial correlation as described below. 

All variables adequately met the assumptions for the chosen analyses. There was considerable attrition of data across the time periods; however, the data did not meet the assumptions for data imputation, so no missing value estimation was performed. Data were analysed using IBM SPSS Statistics 25.

## 3. Results

### 3.1. Baseline Characteristics

Table 1 presents the baseline characteristics of all 54 women randomised to treatment in the original RCT [37]. 

As reported in the follow-up study, inspection of the same characteristics for the sub-set of the 28 women who returned data in the two-year follow-up revealed that the variables appeared to have remained reasonably well balanced between the CBT and TAU groups, with no significant differences between groups. 

### 3.2. Relationship between Social Support and Depression and Anxiety

BDI-II scores, BAI scores, and SPS sub-scales scores across time for the entire sample, irrespective of group membership, are depicted in Table 2.

Table 3 presents the main inferential results from the multivariate multiple regression followed by the results for each dependent measure separately. The multivariate results are the results of the multivariate multiple regressions at each of the four time points, with depression and anxiety as the combined dependent outcomes, and the social support sub-scales as predictors. The univariate results show the results for each dependent measure (i.e., depression or anxiety) separately.

With regard to the multivariate outcomes, there is evidence that the canonical relationship between social support and depression and anxiety combined becomes stronger across the time periods, with the relevant effect size increasing from 0.22 at baseline (T1) to 0.52 at six months post-birth (T3). At twenty four months post-birth (T5), the multivariate relationship is no longer significant, largely due to lack of power associated with the reduced sample size, but the strength of the effect was still 0.38.

With regard to each dependent variable separately, it is clear that the relationship between the combined social support sub-scales and depression was stronger than the relationship between the combined social support sub-scales and anxiety. At all four time points, the combined social support measures were significant predictors of depression, but for anxiety, social support was significant only at nine weeks post-randomisation (T2). There was a notable increase in the strength of the relationship between social support and both outcome measures from baseline (T1) to nine weeks post-randomisation (T2). After that, the relationship tended to level off and marginally weaken.

Examination of individual predictors revealed a consistently strong unique role for the two sub-scales, reassurance of worth and reliable alliance. Reassurance of worth was significant in the model predicting depression at both nine weeks post-randomisation (T2), *p* = 0.005, and six months post-birth (T3), *p* = 0.025, and anxiety at nine weeks post-randomisation (T2), *p* = 0.03. Reliable alliance was also significant in the nine weeks post-randomisation (T2) model predicting both depression, *p* = 0.04, and anxiety, *p* = 0.02. In all instances, these relationships were negative, with higher levels of social support associated with lower levels of depression and anxiety.

### 3.3. Relationship between Social Support and Parenting Stress

There were several strong and significant univariate correlations between parenting stress (parent domain, child domain, and total score) and social support sub-scales at twenty four months post-birth (T5), as illustrated in Table 4.

A multivariate multiple regression failed to find a significant relationship between the social support sub-scales at twenty four months post-birth (T5) and parenting stress, as assessed using parent domain and child domain scores of the PSI, η = 0.18, *F*(12, 18) = 2.05, *p* = 0.08, η_p_^2^ = 0.58.

Despite this non-significant multivariate result, the combined social support sub-scales were found to significantly predict the parent domain score, *p* = 0.01. In that model, no individual social support sub-scales were found to be significant.

### 3.4. Time-Lagged Correlations

Given the potential for social support to have an ongoing impact on depression, anxiety, and parenting stress, a series of time-lagged Pearson correlations were conducted, and are presented in Table 5, Table 6, Table 7 and Table 8. These analyses show the correlation of social support with depression, anxiety, and parenting stress at both the same time point and at one time point removed. For example, social support at baseline is correlated with depression and anxiety at both baseline (T1) and nine weeks post-randomisation (T2) in Table 5. Note that parenting stress, which was measured only at twenty four months (T5), is included along with depression and anxiety in Table 8. 

A general trend that emerged from these analyses was for the correlations between social support and depression and anxiety to be stronger when measured at the same time point than when measured at one time point removed. The notable exception to this was observed at baseline (T1) and nine weeks post-randomisation (T2). The correlations between social support and depression and anxiety were uniformally stronger between social support measured at baseline (T1) and depression and anxiety measured at nine weeks post-randomisation (T2) than between social support and depression and anxiety both measured at baseline (T1).

### 3.5. Role of Social Support in The Relationship Between Depression and Anxiety and Child Outcomes

Given the large number of potential mediation analyses that were available to explore the role of social support in the relationship between depression and anxiety and child outcomes, an initial screening analysis was carried out using partial correlation.

Child outcomes were recorded at nine months and twenty four months post-birth (T4 and T5, respectively). At each time point, Pearson correlation between child outcomes and depression and anxiety revealed a number of significant associations at *p* < 0.05. At nine months post-birth (T4), the following correlations were significant for depression: ASQ total, *r*(28) = 0.46, *p* = 0.01; IBQ duration of orienting, *r*(28) = 0.39, *p* = 0.04; IBQ smiling and laughter, *r*(28) = 0.43, *p* = 0.02; IBQ sadness, *r*(28) = 0.45, *p* = 0.02; and IBQ surgency/extraversion, *r*(28) = 0.39, *p* = 0.04. For anxiety, significant correlations were found for: ASQ total, *r*(26) = 0.41, *p* = 0.04; IBQ activity, *r*(26) = 0.48, *p* = 0.01; IBQ duration of orienting, *r*(26) = 0.49, *p* = 0.01; IBQ smiling and laughter, *r*(26) = 0.61, *p* = 0.001; IBQ cuddliness, *r*(26) = −0.44, *p* = 0.03; IBQ sadness, *r*(26) = 0.51, *p* = 0.008; IBQ approach, *r*(26) = 0.41, *p* = 0.04; IBQ vocal reactivity, *r*(26) = 0.43, *p* = 0.03; and IBQ surgency/extraversion, *r*(26) = 0.56, *p* = 0.003. Overall, relationships between these child outcomes and anxiety were slightly stronger than the relationships with depression, along with more significant associations with depression (five versus nine).

Data for social support were not available at nine months post-birth (T4), so the total social support score at six months post-birth (T3) was partialled out of these correlations in order to screen for possible further mediation analyses.

Even though some of the previously significant correlations became non-significant after partialling out social support, this was largely due to reduced power resulting from decreased sample size. In no instance did the magnitude of the correlations reduce notably, indicating that the relationships between child outcomes and depression and anxiety was largely independent of social support. Given these results, it was felt that further formal mediation analysis was not warranted. 

An equivalent approach was taken for the data at twenty four months post-birth (T5). Initially, only three significant relationships were found between child outcomes and depression and anxiety: Depression was found to correlate significantly with both the Bayley social and emotional sub-scale, *r*(17) = −0.51, *p* = 0.04, and the Bayley SEQ composite score, *r*(17) = −0.51, *p* = 0.04, and anxiety was significantly correlated with the CBCL sleep sub-scale, *r*(27) = 0.43, *p* = 0.02.

As with the results at six months post-birth (T3), partialling out social support at twenty four months post-birth (T5) resulted in no notable change in these values, and a separate analysis partialling out the results of social support at six months post-birth (T3) revealed a similar absence of change. 

## 4. Discussion

### 4.1. Relationship between Social Support, Depression, and Anxiety

The findings provide strong evidence that social support is strongly related to depression from pregnancy to the postpartum in a clinically depressed cohort, consistent with a protective effect of social support. Furthermore, the overall relationship between social support and depression and anxiety combined became stronger across time until six months postpartum. Whilst at each time point, social support was a significant predictor of depression, interestingly, the relationship between social support and anxiety was weaker than for depression. For anxiety, social support was significant only in late pregnancy (nine weeks post-randomisation—T2). The benefit of social support on depression may be best understood by examing which sub-scales appear to provide a protective effect during this vulnerable period in women’s lives.

There are two components of social support that appeared to have a very strong role to play in that relationship: Reassurance of worth and reliable alliance. Higher levels of these aspects of social support were associated with both lower levels of depression and anxiety. Reassurance of worth significantly predicted depression at both nine weeks post-randomisation (T2) and six months post-birth (T3) and anxiety at nine weeks post-randomisation (T2). Reliable alliance also significantly predicted both depression and anxiety at nine weeks post-randomisation (T2), which is the period in late pregnancy close to the infant’s birth. 

‘Reassurance of worth’ refers to relationships in which the woman’s skills and abilities are acknowledged. This supportive reassurance may be of particular importance during the perinatal period due to the significant transition that a woman undergoes with the addition of a new baby to the family. 

‘Reliable alliance’ refers to relationships in which the woman can count on another for assistance under any circumstances. Having others to rely on when needed may be a particularly important protective factor in minimising depression and anxiety during the perinatal period given the often unpredictable nature of many aspects of parenting (e.g., childbirth, child temperament, health, sleep, feeding, etc). Instrumental support (practical help with child care and more general matters), is likely to overlap somewhat with ‘reliable alliance’ (being able to count on others for help), and has also been found to have an inverse relationship with depression [20]. Other forms of social support that may be related to reliable alliance have been reported in other studies; in pregnancy, low partner support [54] and low perceived social support [55] were found to be predictors of subsequent postnatal depression. During the postnatal period, studies have shown that less partner support (partner not available or unreliable for help with childcare) and less support from the woman’s own parents are associated with postnatal depression [56,57,58,59]. 

Interestingly, we did not find ‘guidance’ (relationships with individuals who can provide advice) to be a significant predictor in this study. This is in contrast to the study by Seguin and Potvin [17], which found that the positive impact of informational support was the most significant in protecting against depressive symptoms.

In addition, as described above and shown in Table 3, the association between social support and depression becomes stronger from mid pregnancy (T1—baseline was at around twenty weeks gestation) to six months postpartum (T3). Whilst an increased effect size was not apparent at twenty four months (T5), this might be attributed to the smaller sample size at this time point. Nevertheless, results suggest that the period from late pregnancy (T2—around thirty weeks of pregnancy) to six months (T3) may be a sensitive period for the buffering effect of social support at a time when women are becoming focussed on the approaching birth. Consistent with this observation is that the effects on depression and anxiety were largest and most apparent in late pregnancy (T2—around thirty weeks of pregnancy). Furthermore, in the time-lagged analyses, the correlations between social support and depression and anxiety were strongest when the time lag was from mid to late pregnancy. Again, reassurance of worth and reliable alliance showed strong effects across the time-lagged analyses. Taken together, these results suggest that in the period as childbirth approaches, the impact of prior and current social support on pregnant women’s psychological wellbeing becomes increasingly important.

### 4.2. Social Support and Child Outcomes

From our previous studies with this same cohort [37,41], we know that maternal depression and anxiety impacted on child outcomes, particularly at nine months of age, with numerous associations on the IBQ and ASQ. In the current analyses, it was therefore interesting that the relationships between child outcomes and maternal depression and anxiety were found to be largely independent of social support. That is, there was no evidence that the relationship between depression or anxiety and child outcomes was mediated by social support. 

Whilst one of the few existing studies [29] reported that social support during pregnancy is a protective factor for child development, this was only so in women with socio-demographic risk. Our sample appears to be representative of the Australian childbearing population in terms of age [60], mostly born in Australia, in the middle of the income distribution, and not a particularly high-risk group. Further research is needed to explore other biopsychosocial mechanisms whereby antenatal depression and postnatal depression impact on child development. Currently, there is evidence that there is a direct biological effect in pregnancy [23] and that the quality of the mothering relationship postnatally mediates poor child outcomes [61]. Whilst it appears that, in the current cohort, social support may not have played a key role, Stein and colleagues, in an extensive review, have pinpointed low social support and social disadvantage as correlates of negative child outcomes [61].

### 4.3. Social support and Parenting Stress

Social support at twenty four months also appeared to play a protective role in the experience of parenting stress. Numerous correlations between parenting stress (parent domain, child domain, and total score) were found with social support sub-scales. In addition, the combined social support sub-scales were found to significantly predict the parent domain of the PSI, suggesting that low social support has an impact on the mother’s perception of herself as a parent, e.g., feelings of self-competence, isolation, and role restriction. In the six month to twenty four month time-lagged correlation analyses, there were some similarities to the pattern found for depression and anxiety. Namely, social support and parenting stress were strongly correlated at a single timepoint (twenty four months) and most sub-scales of the SPS at six months postpartum predicted parenting stress at twenty four months (the next subsequent time point where social support was measured).

Our findings were particularly significant in the area of the parent domain rather than in the child domain, suggesting that social support has less of a role in mitigating parenting stress about the child. This is consistent with Cutrona and Troutman [57] who investigated whether social support protects mothers against the stress of daily responsibility for infants. They found that the relationship between social support and postpartum depression was mediated by the mothers’ self-efficacy as a parent. 

### 4.4. Implications for Interventions

This study adds to a considerable literature which describes the importance of social support for perinatal women experiencing depression. Perinatal depression and anxiety have relatively high prevalence [62,63], but both depression and anxiety can be effectively treated [64,65,66]. Early intervention with an effective treatment may not only improve wellbeing, but may also have the potential to reduce the known social and economic costs of mental health problems perinatally [67]. 

Although causality cannot be inferred from our analyses, strengthening social support may alleviate depressive symptoms in perinatal women by developing social support interventions. Currently, cognitive-behavioural therapy (CBT) is considered a best practice intervention for women depressed in the perinatal period; however, CBT approaches generally do not actively foster social support despite accumulating evidence that this may be important for pregnant and new mothers. 

Whilst self-worth (reassurance of worth) may be a target of cognitive techniques, a specific focus on involving social networks to reinforce a cognitive therapy approach is relevant perinatally. Recently, a social support intervention within a CBT framework was trialed successfully in a general population sample who scored low on social support. The intervention was successful at increasing functional, but not structural (e.g., increasing size of social network), support at ten weeks follow-up, suggesting it was useful for stress buffering purposes [68]. Similarly, women at risk of postnatal depression who received social support from their peers were found to be less likely to experience depression [69].

In addition, our finding that reliable alliance is a key aspect of social support in the perinatal period suggests that approaches that integrate couple therapy into perinatal depression treatment [70] may be the interventions of choice. 

In summary, current intervention approaches for perinatal depression may be effectively enhanced by a renewed focus on elements of social support specifically in late pregnancy and the first six months postpartum. 

### 4.5. Quality of Life

Improving social support may play a role not only in women’s depression, but more broadly in terms of her quality of life (QoL). Quality of life is a complex and personal area, affected by many different aspects of health and wellbeing, and over the last decade there has been growing interest in how perinatal women fare. Hill and colleagues [71] created the Maternal Postpartum Quality of Life Questionnaire (MAPP-QoL) for this population with domains including: Psychological/baby, socioeconomic, relational/spouse-partner, relational/family-friends, and health functioning. Whilst postnatal depression has been found to influence all domains of life on QoL measures ([72], social support has been shown to be protective against both postnatal depression and poor QoL [73]. This broader view of a woman’s experience suggests that interventions that take a systemic approach may improve not only depression, but also other important areas of a woman’s life.

### 4.6. Limitations and Other Considerations

It should be noted that the sample size in this study is relatively small (*n* = 54). Further, there was relatively high attrition of the sample with only 28 out of 54 participants returning data at two years. The results should therefore be interpreted with some caution.An important consideration in interpreting findings is that depression is likely to make people appraise their social support as inadequate. Scales such as the SPS measure *perceived* social support and measure functional, not structural support.It has been suggested that an important area of research might be to investigate whether social support can at times have a negative effect, e.g., difficulties with mothers/mother-in-laws may mean that offers of help are not always helpful. Negative effects of social support may explain why a number of studies have not found social support to be an important predictor of depression and/or anxiety during the perinatal period (e.g., Reference [74]).

## 5. Conclusions

Social support appears to be a significant predictor of depression and anxiety from mid-pregnancy to six months postpartum and particularly in late pregnancy. Of the six aspects of social support measured in this study, reassurance of worth and reliable alliance seemed to play the strongest role in this relationships. In addition, this study suggests that social support is a protective factor against parenting stress, specifically in the parent domain, at twenty four months post-birth. Finally, the relationships between depression and anxiety and child outcomes were found to be largely independent of social support in this cohort. Nevertheless, these results support the importance of further research focused on adjusting current treatment approaches for perinatal depression to be more social support-oriented from late pregnancy and across the first six months postpartum. 

## Figures and Tables

**Table 1 ijerph-16-01426-t001:** Participant characteristics at baseline.

	Intervention *n* = 27	Usual Care *n* = 27
**DSM-IV Diagnosis *n* (%)**		
**Major Depression**	21 (77.8)	18 (66.7)
**Minor Depression**	1 (3.7)	4 (14.8)
**Adjustment Disorder with** **mixed depression and anxiety**	5 (18.5)	5 (18.5)
**Age *M* (*SD*)**	30.51 (5.80)	32.80 (5.97)
**Gestational stage in weeks *M * (*SD*)**	21 (6)	20 (8)
**Relationship status *n* (%)**		
**Partner**	25 (92.6)	23 (85.2)
**No Partner**	2 (7.4)	4 (14.8)
**Education *n* (%)**		
**High school only**	3 (11.1)	8 (29.6)
**Certificate level**	6 (22.2)	3 (11.1)
**Diploma level**	5 (18.5)	2 (7.4)
**University degree**	9 (33.3)	10 (37)
**Postgraduate**	4 (14.8)	4 (14.8)
**Number of previous children *n *(%)**		
**0**	14 (51.9)	20 (74)
**≥1**	13 (48.1)	7 (26)
**Born in Australia *n* (%)**	20 (74)	22 (81.5)
**Family Income *n* (%)**		
**up to $80,000**	15 (55.5)	18 (66.6)
**$80,001 or greater**	8 (29.6)	8 (29.6)
**No answer**	4 (14.8)	1 (3.7)

**Table 2 ijerph-16-01426-t002:** Beck Depression Inventory (BDI) scores, Beck Anxiety Inventory (BAI) scores, and social provision scale (SPS) sub-scales scores across time.

	BL (T1) M (SD)	9 weeks PR (T2) M (SD)	6 months Post-birth (T3) M (SD)	* 9 months Post-birth (T4) M (SD)	24 months Post-birth (T5) M (SD)
**BDI**-II	30.77 (9.00)	15.21 (10.29)	14.58 (10.45)	16.34 (14.48)	13.30 (8.29)
**BAI**	21.64 (10.34)	13.81 (10.41)	10.94 (9.59)	10.26 (10.14)	8.07 (6.38)
**Guidance**	12.49 (1.97)	13.63 (2.40)	14.11 (1.84)		13.85 (2.21)
**Reassurance of Worth**	11 (2.15)	12.39 (1.96)	12.91 (2.20)		12.81 (2.40)
**Social Integration**	11.02 (2.69)	12.68 (2.35)	13.48 (2.21)		13.41 (2.24)
**Attachment**	11.45 (2.26)	12.79 (2.91)	13.17 (2.16)		12.81 (2.68)
**Opportunity for Nurturance**	12.79 (2.44)	13.34 (1.62)	14.73 (1.38)		14.56 (1.65)
**Reliable Alliance**	12.62 (2.38)	13.43 (2.34)	13.76 (2.10)		14.07 (2.15)
**Total Score**	71.26 (9.95)	78.59 (10.55)	82.10 (9.80)		81.52 (10.97)

BL, Baseline; PR, Post-randomisation; BDI-II, Beck Depression Inventory; BAI, Beck Anxiety Inventory. * SPS score for the nine-month follow-up were not available.

**Table 3 ijerph-16-01426-t003:** Relationship between social support and depression and anxiety across the four time points.

	Outcome	*p* Value	ηp2
**Multivariate assessment**			
**BL (T1)**	Depression and anxiety	0.03	0.22
**9 weeks PR (T2)**	Depression and anxiety	0.002	0.43
**6 months post-birth (T3)**	Depression and anxiety	0.001	0.52
**24 months post-birth (T5)**	Depression and anxiety	0.08	0.38
**Univariate Assessment**			
**BL (T1)**	Depression	0.009	0.22
**9 weeks PR (T2)**	Depression	0.001	0.53
**6 months post-birth (T3)**	Depression	0.003	0.46
**24 months post-birth (T5)**	Depression	0.03	0.32
**BL (T1)**	Anxiety	0.19	0.06
**9 weeks PR (T2)**	Anxiety	0.002	0.43
**6 months post-birth (T3)**	Anxiety	0.23	0.10
**24 months post-birth (T5)**	Anxiety	0.26	0.09

BL, Baseline; PR, Post-randomisation.

**Table 4 ijerph-16-01426-t004:** Correlations between SPS sub-scales and parenting stress scores at twenty four months post-birth (T5).

		PSI Parent Domain	PSI Child Domain	PSI Total
Guidance	*r*	−0.77	−0.48	−0.73
	*p*	<0.001	0.02	0.001
	*N*	18	25	17
Reassurance of Worth	*r*	−0.70	−0.62	−0.72
	*p*	0.001	0.001	0.001
	*N*	18	25	17
Social Integration	*r*	−0.73	−0.54	−0.70
	*p*	0.001	0.006	0.002
	*N*	18	25	17
Attachment	*r*	−0.71 *	−0.53	−0.74
	*p*	0.001	0.007	0.001
	*N*	18	25	17
Opportunity for Nurturance	*r*	−0.25	−0.27	−0.28
	*p*	0.33	0.20	0.28
	*N*	18	25	17
Reliable Alliance	*r*	−0.67	−0.46	−0.64
	*p*	0.002	0.02	0.006
	*N*	18	25	17

* Correlation is significant at the 0.05 level (2-tailed).

**Table 5 ijerph-16-01426-t005:** Correlations between social support and depression and anxiety at baseline (T1) and nine weeks post-randomisation (T2).

		T1 Depression	T1 Anxiety	T2 Depression	T2 Anxiety
Social Provisions Scale—Guidance Time 1	*r*	−0.08	−0.15	−0.32 *	−0.25
	*p*	0.59	0.28	0.05	0.12
	*N*	51	51	40	41
Social Provisions Scale—Reassurance of Worth Subscale T1	*r*	−0.42 **	−0.31 *	−0.61 **	−0.54 **
	*p*	0.002	0.03	<0.001	<0.001
	*N*	53	53	42	43
Social Provisions Scale—Social Integration T1	*r*	−0.38 **	−0.36 **	−0.53 **	−0.46 **
	*p*	0.006	0.009	<0.001	0.003
	*N*	51	51	40	41
Social Provisions Scale—Attachment T1	*r*	−0.26	−0.27	−0.40 **	−0.28
	*p*	0.06	0.05	0.008	0.07
	*N*	53	53	42	43
Social Provisions Scale—Nurturance T1	*r*	−0.08	−0.17	0.15	<0.01
	*p*	0.59	0.22	0.36	0.99
	*N*	53	53	42	43
Social Provisions Scale—Reliable Alliance T1	*r*	−0.05	−0.26	−0.29	−0.30
	*p*	0.71	0.06	0.07	0.05
	*N*	52	52	41	42
Social Provisions Total Score T1	*r*	−0.33 *	−0.37 **	−0.48 **	−0.44 **
	*p*	0.02	0.008	0.002	0.005
	*N*	50	50	39	40

* Correlation is significant at the 0.05 level (2-tailed). ** Correlation is significant at the 0.01 level (2-tailed).

**Table 6 ijerph-16-01426-t006:** Correlations between social support and depression and anxiety at nine weeks post-randomisation (T2), and six months postpartum (T3).

		T2 Depression	T2 Anxiety	T3 Depression	T3 Anxiety
Social Provisions Scale—Guidance T2	*r*	−0.55 **	−0.57 **	−0.25	0.04
	*p*	<0.001	<0.001	0.19	0.82
	*N*	37	38	30	30
Social Provisions Scale—Reassurance of Worth Subscale T2	*r*	−0.64 **	−0.58 **	−0.58 **	−0.52 **
	*p*	<0.001	<0.001	0.001	0.004
	*N*	35	36	29	29
Social Provisions Scale—Social Integration T2	*r*	−0.59 **	−0.54 **	−0.39 *	−0.27
	*p*	<0.001	0.001	0.04	0.15
	*N*	36	37	29	29
Social Provisions Scale—Attachment T2	*r*	−0.57 **	−0.62 **	−0.23	0.01
	*p*	<0.001	<0.001	0.23	0.95
	*N*	37	38	30	30
Social Provisions Scale—Nurturance T2	*r*	−0.14	−0.22	<0.01	0.07
	*p*	0.43	0.18	0.99	0.72
	*N*	37	38	30	30
Social Provisions Scale—Reliable Alliance T2	*r*	−0.50 **	−0.54 **	−0.36	−0.05
	*p*	0.002	0.001	0.06	0.78
	*N*	36	37	29	29
Social Provisions Total Score T2	*r*	−0.59 **	−0.55 **	−0.42 *	−0.19
	*p*	<0.001	0.001	0.03	0.33
	*N*	33	34	27	27

* Correlation is significant at the 0.05 level (2-tailed). ** Correlation is significant at the 0.01 level (2-tailed).

**Table 7 ijerph-16-01426-t007:** Correlations between social support and depression and anxiety at six months (T3) and nine months (T4) postpartum; Correlations between social support and parenting stress at six months (T3) and twenty four months (T5) postpartum.

		T3 Depression	T3 Anxiety	T4 Depression	T4 Anxiety	T5 PSI Parent	T5 PSI Child	T5 Total PSI
Social Provisions Scale—Guidance T3	*r*	−0.48 **	−0.06	−0.41 *	−0.11	−0.55 *	−0.28	−0.57 *
	*p*	0.005	0.73	0.05	0.64	0.03	0.19	0.03
	*N*	33	33	24	22	16	23	15
Social Provisions Scale—Reassurance of Worth Subscale T3	*r*	−0.69 **	−0.32	−0.60 **	−0.16	−0.71 **	−0.56 **	−0.72 **
	*p*	<0.001	0.07	0.002	0.47	0.002	0.005	0.002
	*N*	33	33	24	22	16	23	15
Social Provisions Scale—Social Integration T3	*r*	−0.60 **	−0.19	−0.48 *	−0.02	−0.68 **	−0.43	−0.66 *
	*p*	<0.001	0.32	0.02	0.94	0.005	0.05	0.01
	*N*	31	31	23	21	15	21	14
Social Provisions Scale—Attachment T3	*r*	−0.56 **	<0.01	−0.49 *	−0.05	−0.59 *	−0.28	−0.58 *
	*p*	0.001	0.97	0.01	0.82	0.02	0.21	0.02
	*N*	33	33	24	22	16	23	15
Social Provisions Scale—Nurturance T3	*r*	−0.05	0.26	0.15	0.28	0.12	0.38	0.26
	*p*	0.80	0.15	0.49	0.22	0.67	0.07	0.35
	*N*	32	32	23	21	16	23	15
Social Provisions Scale—Reliable Alliance T3	*r*	−0.41 *	<0.01	−0.31	−0.04	−0.61 *	−0.20	−0.58 *
	*p*	0.02	0.96	0.14	0.85	0.01	0.36	0.02
	*N*	32	32	24	22	16	23	15
Social Provisions Total Score T3	*r*	−0.64 **	−0.06	−0.52 *	−0.02	−0.67 **	−0.34	−0.65 *
	*p*	<0.001	0.76	0.01	0.93	0.006	0.13	0.01
	*N*	29	29	22	20	15	21	14

* Correlation is significant at the 0.05 level (2-tailed). ** Correlation is significant at the 0.01 level (2-tailed).

**Table 8 ijerph-16-01426-t008:** Correlations between social support, depression, and anxiety, and parenting stress at twenty four months postpartum (T5).

		T5 Depression	T5 Anxiety	T5 PSI Parent	T5 PSI Child	T5 Total PSI
Social Provisions Scale—Guidance T5	*r*	−0.62 **	−0.50 **	−0.77 **	−0.48 *	−0.73 **
	*p*	0.001	0.009	<0.001	0.02	0.001
	*N*	26	26	18	25	17
Social Provisions Scale—Reassurance of Worth Subscale T5	*r*	−0.53 **	−0.19	−0.70 **	−0.62 **	−0.72 **
	*p*	0.005	0.34	0.001	0.001	0.001
	*N*	26	26	18	25	17
Social Provisions Scale—Social Integration T5	*r*	−0.58 **	−0.30	−0.73 **	−0.54 **	−0.70 **
	*p*	0.002	0.13	0.001	0.006	0.002
	*N*	26	26	18	25	17
Social Provisions Scale—Attachment T5	*r*	−0.50 **	−0.39 *	−0.71 **	−0.53 **	−0.74 **
	*p*	0.01	0.05	0.001	0.007	0.001
	*N*	26	26	18	25	17
Social Provisions Scale—Nurturance T5	*r*	−0.38	<0.01	−0.25	−0.27	−0.28
	*p*	0.05	0.99	0.33	0.20	0.28
	*N*	26	26	18	25	17
Social Provisions Scale—Reliable Alliance T5	*r*	−0.60 **	−0.30	−0.67 **	−0.46 *	−0.64 **
	*p*	0.001	0.14	0.002	0.02	0.006
	*N*	26	26	18	25	17
Social Provisions Total Score T5	*r*	−0.66 **	−0.36	−0.79 **	−0.60 **	−0.77 **
	*p*	<0.001	0.07	<0.001	0.002	<0.001
	*N*	26	26	18	25	17

* Correlation is significant at the 0.05 level (2-tailed). ** Correlation is significant at the 0.01 level (2-tailed).

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
