# Peer review of "Social Support—A Protective Factor for Depressed Perinatal Women?"

_ijerph, 2019, doi:10.3390/ijerph16081426_

Round 1
Reviewer 1 Report
I am reviewing the paper entitled “Social Support—A Protective Factor for Perinatal Depression” for the International Journal of Environmental Research and Public Health. The authors need to refocus the statistics that they conduct and set up those statistics in the introduction. I am going to lay out the problems with the current analyses and solutions to these problems and then I will point out grammar errors in the first three sections of the paper. I admit that I did not read the Discussion because it will radically change with the change in statistics.
The authors need to remember that they need to set up (justify the importance and reason) in the introduction the theory and literature for all the analyses that will be conducted in the results as those results are the point of the paper. With all the statistics currently in the paper, the introduction would demand 15 to 25 pages to sufficiently set up the 100 or so relations in tables and text with literature to justify their exploration. The analyses, as they currently stand in the paper, often feel like a shotgun approach with many relations that are presented inconsistently and sporadically in a way that does not tell a central story. The authors also need to realize that they have a small sample, likely because the sample is difficult to collect, so they cannot conduct moderation analyses because they typically demand a very large sample and/or a large effect to be significant. Given the small sample, if the authors have a strong theoretical reason for a large effect, they need to set up that theory in the introduction.
In addition, the authors say that anxiety and depression are the dependent variables, but then they examine whether social support predicts them as some sort of combined measure in multiple regression? This part of the paper was extremely confusing. Did the authors try to predict social support as the criterion variable with anxiety and depression as the predictors using standard regression? I do not understand this analysis and it does not add anything to our knowledge. The authors said that anxiety, depression, and stress were criterion variables, not predictors. Any combination of variables into a single criterion would need to be justified in the introduction but nothing really comes from those analyses beyond the results of depression alone (except for one time period), so the overall outcome for these additional analyses is not worth the work that would need to be provided to justify them and they are confusing.
As for the presentation of the statistics, the authors should provide four correlation tables with each table targeting a time period. In each table, the authors should place the correlations between social support, depression, anxiety, and stress, as well as the child outcomes. The authors should not try to examine social support as a mediator because social support is much more likely to drive anxiety, depression, and stress than the opposite is to be true. Although anxiety, depression, and stress could reduce the amount of social support that one receives, these possibilities are much less likely than social support being able to reduce anxiety, depression, and stress. Based on that logic, the authors could examine whether anxiety, depression, and stress mediate the relation of social support and child outcomes. Again, the authors would need to set up these possibilities with literature in the introduction.
Of course, the authors do not need to simply rely on logic as they have the data to examine whether anxiety, depression, and stress lead to social support or vice versa, because they collect data across time. The authors could examine many possibilities, such as the ability of anxiety, depression, and stress at T1 to predict social support at T2, T3, and T5 and they could examine the ability of social support at T1 to predict anxiety, depression, and stress at T2, T3, and T5 in cross lagged panel analyses. However, the authors should probably focus on relations separated by one time period. For example, the authors could examine anxiety, depression, and stress at T1 as predictors of social support at T2 and they could examine social support at T1 as a predictor of anxiety, depression, and stress at T2. The cross lagged correlation that is stronger determines the more likely predictive relation. The same analyses could be conducted across T2 and T3, as well as T3 and T5 and the same logic applies. The logic of the limitation in time period comparison is that one does not need to conduct and present every possible relation to establish directionality/time precedence (one of three components for causality). The fact that predictor variables at T1 could predict criterion variables at T5 would be impressive, but this outcome is not expected, and it is not necessary to establish directionality. In addition, the number of relations in the current study should be limited for two reasons: literature and theory are needed to set up each relation in the introduction and the overall alpha level for relations in the current study is being inflated with each additional test.
As stated previously, the sample is small, but it is large enough to allow an examination of correlations of social support with anxiety, depression, and stress as well as the child outcomes, so anxiety, depression, and stress can be examined as mediators of the relation between social support and child outcomes. The authors do not need to examine anxiety, depression, and stress as mediators of the relation between social support and child outcomes to publish this paper, but these examinations are the most logical and straightforward given the method used and the results found by the authors. Of course, the authors would need to set up these relations in the introduction with the necessary literature. If the authors examine whether both social support as well as anxiety, depression, and stress predict future child outcomes (establishing directionality), they should do so in a manner like the one described previously. Specifically, the relations could be examined between the predictors at T1 and the child outcomes at T2, and the relations could be examined between the predictors at T2 and the child outcomes at T3, etc., separating predictors and criterion variables by 1 time period to reduce the number of analyses to the most necessary and important ones. After establishing all the necessary relations via correlations in the previously mentioned tables, showing that social support predicts anxiety, depression, stress and the child outcomes via the cross lagged correlations, and showing that anxiety, depression, and stress predict the child outcomes via cross lagged correlations, the authors could test for mediation. Specifically, the authors would need to remove (partial out) anxiety, depression, and stress from the relations of social support and child outcomes.
Mediation is simple in that the potential mediator is removed from the relation of the predictor and the criterion, and mediation is achieved if the relation between the predictor and criterion is reduced. However, the mediation analyses in the current study could become quite complex due to time period and all the measures. In other words, the mediation analyses could merely be conducted with all the variables at T5, which would be simple. Alternatively, social support at T1 could predict mediator variables (anxiety, depression, and stress) at T1 or T2, which would predict child outcomes at T5. Similarly, social support at T2 could be used to predict mediator variables at T2 or T3, which would predict child outcomes at T5. By extension, social support at T3 could be used to predict mediator variables at T3 or T5, which would predict child outcomes at T5. Finally, social support at T3 could be used to predict mediator variables at T5, which would predict child outcomes at T5. Given the large number of analyses that would need to be conducted across time, I believe that the authors should simply conduct mediation analyses with the variables at T5 after they have established directionality for 1) social support predicting both the mediator variables and the child outcomes and 2) the mediator variables predicting the child outcomes. Again, any and all mediation analyses must be set up in the introduction, meaning that they need to be justified with supporting literature.
I will now tackle the grammar issues in the beginning of the paper. At the end of the third sentence of the abstract, the authors put two periods. In the sentence starting “There was a strong”, the authors should continue “relationship, particularly in late pregnancy, between two…” In the second paragraph of the introduction, the authors say “less social support” and “more depressed”, but less and more than what? The authors can simply say “low social support report high depression”. Three sentences later, the authors say “to1-year” together and then they should say “latter study” later in that sentence. In the next paragraph in the third sentence, the authors should put a comma after “6 dimensions” and before “which”. The next sentence should end “another individual.” In the next sentence, the authors should delete the “and” after “pregnancy” and before “over”. In the third sentence of the next paragraph, the authors say “wasa”. In the next sentence, the authors should say “Of the few studies conducted on this topic” and they should put commas around “in turn” later in that sentence. The next sentence should say “Women experiencing low social support reported low birth weight …poor labour progress…low Apgar scores.” The next sentence should say “women experiencing low social support …were born early and provided lower…” The first sentence in the next paragraph should put a comma before “were towfold” and say “were two fold.” The next sentence should start “First, the current study examined….” The sentence fragment starting “however” after the semicolon should start anew with “However,” and later say “we will also evaluate the relationship…” and put a comma after “stress” and before “which” in the last sentence of that paragraph. The next paragraph should start “Second,” and the last sentence with the semicolon and “here” should end the sentence with a period instead of a semicolon and start “In the current study,”. The next sentence should start “We hypothesized…depression and anxiety, and it influences antenatal…depression during child development.”
On page 3 in the second sentence, the authors should say “or to TAU, which meant that women”. The measures section should put a comma after “PSI” and before “which”. In the section of 2.3.1, the authors need two sentences. Two sentences are needed in every paragraph and section. In the BDI and BAI sections, the authors need to describe the scale in the form of 0 (no or low) to 3 (high) or the specific form that they need for the scale. In the second sentence of the SPS section, the authors should replace “are:” with “include:”. In the section 2.3.3.1., the authors should end the paragraph with “follow-up test session.” The next two sections should be ended the same way because they are ending with a preposition otherwise. In section 2.3.3.3., the authors should put “and” between “affect,” and “interaction”. In section 2.3.3.4, the authors should end “The test was administered and rated by a clinician.” On page 5, the authors should say “To achieve this outcome,”. The first sentence of the next paragraph should say “social support and depression, anxiety, and stress”. The next sentence should start “The depression, anxiety, and stress scores…dependent outcomes, with the individual and combined social support scores as the predictors.” The authors should say “4” instead of “four” in the next sentence, they should delete the next sentence and say “The analysis of the role”. In the second sentence of the next paragraph, the authors should say “considerable” rather than “considerably”.
The authors can certainly put T1-T5 in the tables, but they also explain them. However, the authors should always simply describe the time periods with words in text and in statements and titles in the tables. A conclusion section is a conclusion, meaning that it must be the last section, not the penultimate section. The authors should move the quality of life section before the conclusion section to make the conclusion section a conclusion section.
Author Response
Response to Reviewer 1 Comments
Point 1: The authors need to remember that they need to set up (justify the importance and reason) in the introduction the theory and literature for all the analyses that will be conducted in the results as those results are the point of the paper. With all the statistics currently in the paper, the introduction would demand 15 to 25 pages to sufficiently set up the 100 or so relations in tables and text with literature to justify their exploration. The analyses, as they currently stand in the paper, often feel like a shotgun approach with many relations that are presented inconsistently and sporadically in a way that does not tell a central story.
Response 1: In responding to this comment from Reviewer 1 we have re-focused the final paragraphs of the Introduction to more closely reflect and map to the analyses that are carried out and to better make the connection between existing theory and literature and our research questions. The revised section now reads “Much of the social support research in the perinatal period has focussed on links with depression. However, studies have also begun to examine the relationship between social support and perinatal anxiety [8,38] alongside depression as a measure of psychological wellbeing in the transition to parenthood [39]. In addition, we measured parenting stress, which is commonly elevated in women experiencing postnatal depression and anxiety [40] [41].
The primary aim was therefore to explore the pattern of longitudinal relationships between social support and psychological wellbeing (i.e., depression, anxiety, and parental stress) in this cohort. Two main methodological approaches were employed: (1) multivariate analyses were used to evaluate the canonical relationships between multiple measures of social support (i.e., the sub-scales of our social support measure) and our measures of psychological wellbeing (i.e., anxiety and depression). Mulitvariate multiple regression conducted at four time points was the analysis of choice here. (2) Time-lagged correlations were used to explore the possible long-term impact of social support on psychological wellbeing. For these analyses, the focus was on correlations between social support at one time point and psychological wellbeing at a subsequent point.
A secondary aim concerned the possible role of social support as a mediator of the relationship between parental psychological wellbeing and child behaviour outcomes. Partial correlation was used to provide a preliminary investigation of this relationship.
In summary we hypothesised that social support plays a protective role against perinatal depression and anxiety, and influences the effect of antenatal and postnatal depression on child development.”
We believe that this revision addresses the thrust of Reviewer’s comment as regards justifying the importance and reasons for the analyses we have conducted.
Point 2: The authors also need to realize that they have a small sample, likely because the sample is difficult to collect, so they cannot conduct moderation analyses because they typically demand a very large sample and/or a large effect to be significant. Given the small sample, if the authors have a strong theoretical reason for a large effect, they need to set up that theory in the introduction.
Response 2: This is a fair comment. Since the moderation analyses that were conducted did not generate any results of statistical or theoretical interest, the paper loses nothing by removing these analyses. Hence, all reference to moderation analysis has been removed from the manuscript. We hope this addresses the Reviewer’s comment.
Point 3: The authors say that anxiety and depression are the dependent variables, but then they examine whether social support predicts them as some sort of combined measure in multiple regression? This part of the paper was extremely confusing. Did the authors try to predict social support as the criterion variable with anxiety and depression as the predictors using standard regression? I do not understand this analysis and it does not add anything to our knowledge. The authors said that anxiety, depression, and stress were criterion variables, not predictors. Any combination of variables into a single criterion would need to be justified in the introduction but nothing really comes from those analyses beyond the results of depression alone (except for one time period), so the overall outcome for these additional analyses is not worth the work that would need to be provided to justify them and they are confusing.
Response 3: We are happy take on board these comments of the Reviewer, and welcome the opportunity to clarify these analyses for the readership. The theoretical basis for these analyses is that social support is a predictor of depression and anxiety; specifically, that level of social support directly impacts levels of depression and anxiety. From our perspective, we see the relationship as unidirectional. The chosen analyses reflect this with depression and anxiety as the dependent outcomes and social support as the predictors. The chosen analysis of multivariate multiple regression, we would argue, is a direct and appropriate test of our main hypotheses. Depression and anxiety are reflections of the underlying construct of psychological wellness, so combining them into a single multivariate outcome is clinically, theoretically, and statistically sound. Similarly, social support is a single construct reflected by a number of sub-scales in the social provisions scale, so a multiple regression analysis answers an important question of interest. With a multivariate relationship established, each dependent outcome can be examined individually, and then the unique role of each of the social support variables can be considered. These analyses are somewhat similar to a canonical correlation, but are more focussed, and clearly distinguish between the outcome variables and the predictors. Thus, we believe that these analyses are central to our core research questions and should remain. However, in light of the Reviewer’s comments we acknowledge that these considerations were not clearly communicated to the reader in our previous version. Accordingly, we have amended the wording to try to clarify these aspects of our approach and to avoid any confusion for the reader. We hope that these revisions are acceptable.
Point 4: As for the presentation of the statistics, the authors should provide four correlation tables with each table targeting a time period. In each table, the authors should place the correlations between social support, depression, anxiety, and stress, as well as the child outcomes.
Response 4: Please see the response to Point 5, which we believe also addresses this point.
Point 5: The authors should not try to examine social support as a mediator because social support is much more likely to drive anxiety, depression, and stress than the opposite is to be true. Although anxiety, depression, and stress could reduce the amount of social support that one receives, these possibilities are much less likely than social support being able to reduce anxiety, depression, and stress. Based on that logic, the authors could examine whether anxiety, depression, and stress mediate the relation of social support and child outcomes. Again, the authors would need to set up these possibilities with literature in the introduction.
Of course, the authors do not need to simply rely on logic as they have the data to examine whether anxiety, depression, and stress lead to social support or vice versa, because they collect data across time. The authors could examine many possibilities, such as the ability of anxiety, depression, and stress at T1 to predict social support at T2, T3, and T5 and they could examine the ability of social support at T1 to predict anxiety, depression, and stress at T2, T3, and T5 in cross lagged panel analyses. However, the authors should probably focus on relations separated by one time period. For example, the authors could examine anxiety, depression, and stress at T1 as predictors of social support at T2 and they could examine social support at T1 as a predictor of anxiety, depression, and stress at T2. The cross lagged correlation that is stronger determines the more likely predictive relation. The same analyses could be conducted across T2 and T3, as well as T3 and T5 and the same logic applies. The logic of the limitation in time period comparison is that one does not need to conduct and present every possible relation to establish directionality/time precedence (one of three components for causality). The fact that predictor variables at T1 could predict criterion variables at T5 would be impressive, but this outcome is not expected, and it is not necessary to establish directionality. In addition, the number of relations in the current study should be limited for two reasons: literature and theory are needed to set up each relation in the introduction and the overall alpha level for relations in the current study is being inflated with each additional test.
Response 5: We appreciate the helpful suggestion to provide time-lagged correlational analyses, and we agree that this will provide a worthwhile complement to the analyses that have already been reported. We have included correlation matrices that provide results of these analyses, with some associated commentary added to the revised text. We hope this revision sufficiently address both Point 4 and Point 5.
Point 6: As stated previously, the sample is small, but it is large enough to allow an examination of correlations of social support with anxiety, depression, and stress as well as the child outcomes, so anxiety, depression, and stress can be examined as mediators of the relation between social support and child outcomes. The authors do not need to examine anxiety, depression, and stress as mediators of the relation between social support and child outcomes to publish this paper, but these examinations are the most logical and straightforward given the method used and the results found by the authors. Of course, the authors would need to set up these relations in the introduction with the necessary literature. If the authors examine whether both social support as well as anxiety, depression, and stress predict future child outcomes (establishing directionality), they should do so in a manner like the one described previously. Specifically, the relations could be examined between the predictors at T1 and the child outcomes at T2, and the relations could be examined between the predictors at T2 and the child outcomes at T3, etc., separating predictors and criterion variables by 1 time period to reduce the number of analyses to the most necessary and important ones. After establishing all the necessary relations via correlations in the previously mentioned tables, showing that social support predicts anxiety, depression, stress and the child outcomes via the cross lagged correlations, and showing that anxiety, depression, and stress predict the child outcomes via cross lagged correlations, the authors could test for mediation. Specifically, the authors would need to remove (partial out) anxiety, depression, and stress from the relations of social support and child outcomes. Mediation is simple in that the potential mediator is removed from the relation of the predictor and the criterion, and mediation is achieved if the relation between the predictor and criterion is reduced. However, the mediation analyses in the current study could become quite complex due to time period and all the measures. In other words, the mediation analyses could merely be conducted with all the variables at T5, which would be simple. Alternatively, social support at T1 could predict mediator variables (anxiety, depression, and stress) at T1 or T2, which would predict child outcomes at T5. Similarly, social support at T2 could be used to predict mediator variables at T2 or T3, which would predict child outcomes at T5. By extension, social support at T3 could be used to predict mediator variables at T3 or T5, which would predict child outcomes at T5. Finally, social support at T3 could be used to predict mediator variables at T5, which would predict child outcomes at T5. Given the large number of analyses that would need to be conducted across time, I believe that the authors should simply conduct mediation analyses with the variables at T5 after they have established directionality for 1) social support predicting both the mediator variables and the child outcomes and 2) the mediator variables predicting the child outcomes. Again, any and all mediation analyses must be set up in the introduction, meaning that they need to be justified with supporting literature.
Response 6: We appreciate the detailed advice and specific suggestions the Reviewer has provided on this point. The following quote from the Reviewer’s comments is most salient to our response, “I believe that the authors should simply conduct mediation analyses with the variables at T5 after they have established directionality for 1) social support predicting both the mediator variables and the child outcomes and 2) the mediator variables predicting the child outcomes.” This is effectively what we have done, except we conducted the analyses at time 4 and time 5. Our goal here was to see whether there was any indication that social support mediated the relationship between child outcomes and depression and anxiety; that is our only mediational relationship of theoretical interest. Child outcomes were measured only at times 4 and 5; hence, that was our focus. Rather than conduct a large number of specific mediation analysis using the Hayes and Preacher approach, which, as the Reviewer indicates, would have necessitated establishing initial directionality, we decided to run partial correlation as an initial screening to see whether there was any justification for more detailed mediation analyses. Unlike formal mediation testing, partial correlation does not require initially establishing significant bivariate pathways as a pre-condition. These analyses revealed no notable impact of social support on the relationship between child outcomes and depression and anxiety, so no further mediation analysis was carried out. We therefore feel that this analysis is essentially appropriate as presented, although for the final analysis, we have now included the results of social support at T3 being partialled out of the relationships at T5. Finally, while we considered simply removing all of these analysis, as was mentioned by the Reviewer early in their review of this point, our preference is to retain them in the manuscript. These results are contrary to results reported in previous research by other teams, and so provide what we feel are interesting findings. We hope that this explanation of our thinking and the small adjustment outlined above are sufficient.
Point 7: At the end of the third sentence of the abstract, the authors put two periods. In the sentence starting “There was a strong”, the authors should continue “relationship, particularly in late pregnancy, between two…” In the second paragraph of the introduction, the authors say “less social support” and “more depressed”, but less and more than what? The authors can simply say “low social support report high depression”. Three sentences later, the authors say “to1-year” together and then they should say “latter study” later in that sentence. In the next paragraph in the third sentence, the authors should put a comma after “6 dimensions” and before “which”. The next sentence should end “another individual.” In the next sentence, the authors should delete the “and” after “pregnancy” and before “over”. In the third sentence of the next paragraph, the authors say “wasa”. In the next sentence, the authors should say “Of the few studies conducted on this topic” and they should put commas around “in turn” later in that sentence. The next sentence should say “Women experiencing low social support reported low birth weight …poor labour progress…low Apgar scores.” The next sentence should say “women experiencing low social support …were born early and provided lower…” The first sentence in the next paragraph should put a comma before “were towfold” and say “were two fold.” The next sentence should start “First, the current study examined….” The sentence fragment starting “however” after the semicolon should start anew with “However,” and later say “we will also evaluate the relationship…” and put a comma after “stress” and before “which” in the last sentence of that paragraph. The next paragraph should start “Second,” and the last sentence with the semicolon and “here” should end the sentence with a period instead of a semicolon and start “In the current study,”. The next sentence should start “We hypothesized…depression and anxiety, and it influences antenatal…depression during child development.” On page 3 in the second sentence, the authors should say “or to TAU, which meant that women”. The measures section should put a comma after “PSI” and before “which”. In the section of 2.3.1, the authors need two sentences. Two sentences are needed in every paragraph and section. In the BDI and BAI sections, the authors need to describe the scale in the form of 0 (no or low) to 3 (high) or the specific form that they need for the scale. In the second sentence of the SPS section, the authors should replace “are:” with “include:”. In the section 2.3.3.1., the authors should end the paragraph with “follow-up test session.” The next two sections should be ended the same way because they are ending with a preposition otherwise. In section 2.3.3.3., the authors should put “and” between “affect,” and “interaction”. In section 2.3.3.4, the authors should end “The test was administered and rated by a clinician.” On page 5, the authors should say “To achieve this outcome,”. The first sentence of the next paragraph should say “social support and depression, anxiety, and stress”. The next sentence should start “The depression, anxiety, and stress scores…dependent outcomes, with the individual and combined social support scores as the predictors.” The authors should say “4” instead of “four” in the next sentence, they should delete the next sentence and say “The analysis of the role”. In the second sentence of the next paragraph, the authors should say “considerable” rather than “considerably”.
Response 7: Thank you for all of these suggestions. All of the typographical errors, and grammatical, formatting and stylistic suggestions made by the Reviewer have now been addressed in the revised manuscript. Please note that some suggestions for re-wording made by the Reviewer under Point 7 have been superseded by the requested re-writing of material in the Introduction (see Response to Point 1). While we have not always adopted the exact re-wording suggested by the Reviewer, we have acted on the substance of all suggestions in the most appropriate language (for example, the suggested phrase “test sessions” would not exactly be an accurate reflection of our methods of follow-up data collection, but we have introduced the term “evaluation” which is accurate/appropriate and addresses the Reviewer’s basic point). We hope that these revisions are satisfactory.
Point 8: The authors can certainly put T1-T5 in the tables, but they also explain them. However, the authors should always simply describe the time periods with words in text and in statements and titles in the tables.
Response 8: This is a very good suggestion and makes the manuscript easier to read. As suggested we have described the time periods with words when referred to in the text (followed by T1-T5 in parenthesis) and also in the titles of tables. We hope this sufficiently addresses the Reviewer’s comment.
Point 9: A conclusion section is a conclusion, meaning that it must be the last section, not the penultimate section. The authors should move the quality of life section before the conclusion section to make the conclusion section a conclusion section.
Response 9: We agree with the Reviewer. The Quality of life section has now been moved to the section immediately before Limitations as we feel it gives a better flow to the text and the Conclusions is now the final section of the paper. We hope this is satisfactory.

Reviewer 2 Report
This is a useful paper in need of minor revision.
Title
The title needs to indicate that the focus is on a treatment cohort, not pregnant women in general.
Abstract
The number of participants should be stated.
1. Introduction
'Social support has been consistently shown to play a protective role against depression in the general population [1-3].'
I agree with the point in general, but 'consistently' is an overstatement. Social support can also have negative effects (as noted in 4.4 Limitations), including being a risk factor for depression. For example:
Shannon Ang, Rahul Malhotra. Association of received social support with depressive symptoms among older males and females in Singapore: Is personal mastery an inconsistent mediator? Social Science & Medicine, 2016; 153: 165 DOI: 10.1016/j.socscimed.2016.02.019
(discussed at https://www.sciencedaily.com/releases/2016/03/160304121645.htm)
2.4. Statistical Analysis
Second sentence: 'these analyses' is unclear – the analyses in this study?
What statistical software was used?
4.4. Limitations
The small sample size should be acknowledged.
So should the high attrition – only 28/54 returned data in the 2-year follow-up (page 5).
5.1. Implications for interventions
Paragraph 1, page 10:
'Given the relatively high prevalence of perinatal depression and anxiety [64], it is important to intervene early to not only improve wellbeing, but to reduce the known social and economic costs of mental health problems perinatally' [65].
65. Bauer, A.; Parsonage, M.; Knapp, M.; Iemmi, V.; Adelaja, B. The costs of perinatal mental health problems; London School of Economics and the Centre for Mental Health: London, 2014.
Firstly, cited reference 65, Bauer et al. (2014) does not provide evidence that early intervention actually does improve wellbeing and reduces the social and economic costs of mental health problems perinatally. Instead, it presents a case that it could do so – it is really wishful thinking dressed up with economic detail.
Either a different reference or references, providing evidence that early intervention improves wellbeing and reduces social and economic costs, should be cited, or the statement should be modified so that it accurately reflects Bauer et al.
Secondly, the high prevalence of a disorder is not an adequate reason to intervene, unless there is evidence that intervention is effective.
Thirdly, the evidence presented in this paper comes from a treatment cohort, and would not necessarily generalise to prevalent cases in the population.
Spelling, grammar, formatting
Spelling mistakes:
· 'considerably' should be 'considerable' (2.4 Statistical Analysis, last paragraph)
· 'andmost' (page 9, second-last paragraph)
· 'enhaced' (page 11, last paragraph)
There are some ungrammatical sentences, including the first two sentences in 1. Introduction (page 1):
· 'Social support in the before and after childbirth….'
· 'Currently there is lack….'
· 'The analysis of role that social support…' (2.4 Statistical Analysis, p. 5)
There are some excessively long paragraphs at the top of page 2, and page 9 paragraphs 3 and 6 (continued on page 10), and the 5.1 Implications paragraph (pages 10-11).
Author Response
Response to Reviewer 2 Comments
Point 1: Title
The title needs to indicate that the focus is on a treatment cohort, not pregnant women in general.
Response 1: This is a good suggestion. We have revised the title, which now reads: “Social Support – A Protective Factor for Depressed Perinatal Women?” We think this better indicates to the reader that the study reports on a clinically depressed sample and not on pregnant women in general. We have also adjusted the Abstract to better reflect this feature of the study. We hope that these changes are acceptable.
Point 2: Abstract
The number of participants should be stated.
Response 2: Thank you for this comment. This information has now been added to the Abstract. The relevant sentence now reads “The present study followed up a cohort of depressed women (n=54) from a randomised controlled trial” We hope that this addresses the Reviewer’s comment.
Point 3: 1. Introduction
'Social support has been consistently shown to play a protective role against depression in the general population [1-3].'
I agree with the point in general, but 'consistently' is an overstatement. Social support can also have negative effects (as noted in 4.4 Limitations), including being a risk factor for depression. For example:
Shannon Ang, Rahul Malhotra. Association of received social support with depressive symptoms among older males and females in Singapore: Is personal mastery an inconsistent mediator? Social Science & Medicine, 2016; 153: 165 DOI: 10.1016/j.socscimed.2016.02.019
(discussed at https://www.sciencedaily.com/releases/2016/03/160304121645.htm)
Response 3: We agree that we should not overstate the evidence, given that whilst many studies show strong association between depression and social support, this is not always the case. The word “consistently” has therefore been taken out. We hope this is a sufficient revision.
Point 4: 2.4. Statistical Analysis
Second sentence: 'these analyses' is unclear – the analyses in this study?
What statistical software was used?
Response 4: Thank you for these suggestions. As suggested, the phrase ‘these analyses’ has now been changed to ‘the analyses in this study’. The sentence “Data were analysed using IBM SPSS Statistics 25” has now been added to the Statistical Analysis section. We hope this is satisfactory.
Point 5: 4.4. Limitations
The small sample size should be acknowledged.
So should the high attrition – only 28/54 returned data in the 2-year follow-up (page 5).
Response 5: We agree with the Reviewer and have adjusted the text accordingly. These issues have now been acknowledged in a new paragraph in the Limitations section. The new text reads “It should be noted that the sample size in this study is relatively small (n=54). Further, there was relatively high attrition of the sample with only 28 out of 54 participants returning data at 2-years. The results should therefore be interpreted with some caution”. We hope this is satisfactory and addresses the Reviewer’s point.
Point 6: 5.1. Implications for interventions
Paragraph 1, page 10:
'Given the relatively high prevalence of perinatal depression and anxiety [64], it is important to intervene early to not only improve wellbeing, but to reduce the known social and economic costs of mental health problems perinatally' [65].
65. Bauer, A.; Parsonage, M.; Knapp, M.; Iemmi, V.; Adelaja, B. The costs of perinatal mental health problems; London School of Economics and the Centre for Mental Health: London, 2014.
Firstly, cited reference 65, Bauer et al. (2014) does not provide evidence that early intervention actually does improve wellbeing and reduces the social and economic costs of mental health problems perinatally. Instead, it presents a case that it could do so – it is really wishful thinking dressed up with economic detail.
Either a different reference or references, providing evidence that early intervention improves wellbeing and reduces social and economic costs, should be cited, or the statement should be modified so that it accurately reflects Bauer et al.
Secondly, the high prevalence of a disorder is not an adequate reason to intervene, unless there is evidence that intervention is effective.
Thirdly, the evidence presented in this paper comes from a treatment cohort, and would not necessarily generalise to prevalent cases in the population.
Response 6: This is a good point and we agree that the text needs revised here for clarity and accuracy We have therefore moderated the language in this section to be phrased more carefully. We have provided appropriate references to substantiate each of the separate, but connected, statements made here. We have clarified that intervention has potential to reduce economic costs, rather than stating (as we did previously) that it will reduce economic costs. The revised text now reads: “Perinatal depression and anxiety have relatively high prevalence, but both depression and anxiety can be effectively treated. Early intervention with an effective treatment may not only improve wellbeing, but may also have the potential to reduce the known social and economic costs of mental health problems perinatally”. We hope that this addresses the Reviewer’s comment sufficiently.
Point 7: Spelling, grammar, formatting
Spelling mistakes:
· 'considerably' should be 'considerable' (2.4 Statistical Analysis, last paragraph)
· 'andmost' (page 9, second-last paragraph)
· 'enhaced' (page 11, last paragraph)
Response 7: Thank you. These errors have now been rectified.
Point 8: There are some ungrammatical sentences, including the first two sentences in 1. Introduction (page 1):
· 'Social support in the before and after childbirth….'
· 'Currently there is lack….'
· 'The analysis of role that social support…' (2.4 Statistical Analysis, p. 5)
Response 8: Thank you. These have now been corrected.
Point 9: There are some excessively long paragraphs at the top of page 2, and page 9 paragraphs 3 and 6 (continued on page 10), and the 5.1 Implications paragraph (pages 10-11).
Response 9: We agree that some passages are too long. We have edited the manuscript throughout, with an eye to introducing paragraph breaks where appropriate. We think this has helped with the readability of the manuscript and hope that this is satisfactory.
